# Ocular surface manifestations of coronavirus disease 2019 (COVID-19): A systematic review and meta-analysis

Kanika Aggarwal[1], Aniruddha Agarwal[1], Nishant Jaiswal[2], Neha Dahiya[3], Alka Ahuja[1], Sarakshi Mahajan[3], Louis Tong[4,5,6,7], Mona Duggal[1], Meenu Singh[2], Rupesh Agrawal[4,5,8,9], Vishali Gupta[1] *

1 Advanced Eye Centre, Department of Ophthalmology, Postgraduate Institute of Medical Education and Research (PGIMER), Chandigarh, India, 2 Department of Pediatrics, Postgraduate Institute of Medical Education and Research (PGIMER), Chandigarh, India, 3 School of Medicine, St Joseph Mercy Hospital, Oakland, Pontiac, Michigan, United States of America, 4 Singapore National Eye Centre, Singapore, Singapore, 5 Singapore Eye Research Institute, Singapore, Singapore, 6 Eye-Academic Clinical Program, Duke-National University of Singapore (NUS) Medical School, Singapore, Singapore, 7 Yong Loo Lin School of Medicine, National University of Singapore, Singapore, Singapore, 8 Moorfields Eye Hospital, NHS Foundation Trust, London, United Kingdom, 9 National Healthcare Group Eye Institute, Tan Tock Seng Hospital, Singapore, Singapore

* vishalisara@yahoo.co.in, vishalisara@gmail.com

**Data Availability Statement:** All relevant data are within the manuscript and its Supporting Information files.

## Abstract

### Purpose

This study was performed to determine the occurrence of ocular surface manifestations in patients diagnosed with coronavirus disease 2019 (COVID-19) due to severe acute respiratory syndrome coronavirus 2 (SARS-CoV-2).

### Methods

A systematic search of electronic databases i.e. PubMed, Web of Science, CINAHL, OVID and Google scholar was performed using a comprehensive search strategy. The searches were current through 31st May 2020. Pooled data from cross-sectional studies was used for meta-analysis and a narrative synthesis was conducted for studies where a meta-analysis was not feasible.

### Results

A total of 16 studies reporting 2347 confirmed COVID-19 cases were included. Pooled data showed that 11.64% of COVID-19 patients had ocular surface manifestations. Ocular pain (31.2%), discharge (19.2%), redness (10.8%), and follicular conjunctivitis (7.7%) were the main features. 6.9% patients with ocular manifestations had severe pneumonia. Viral RNA was detected from the ocular specimens in 3.5% patients.

### Conclusion

The most common reported ocular presentations of COVID-19 included ocular pain, redness, discharge, and follicular conjunctivitis. A small proportion of patients had viral RNA in

**Funding:** The authors have received no specific funding for this work.

**Competing interests:** The authors have declared that no competing interests exist.

their conjunctival/tear samples. The available studies show significant publication bias and heterogeneity. Prospective studies with methodical collection and data reporting are needed for evaluation of ocular involvement in COVID-19.

## Introduction

The outbreak of coronavirus disease 2019 (COVID-19) started in Wuhan, China in December 2019 and rapidly spread globally. It was declared a public health emergency of international concern on 30th January 2020 and in March 2020, it was labelled as a pandemic by the World Health Organization (WHO) [1]. More than 8 million confirmed cases of COVID-19 and nearly 450,000 deaths worldwide have been reported as of 20[th] June 2020 [1, 2]. The causative pathogen of this potentially fatal disease has been named severe acute respiratory syndrome coronavirus2 (SARS-CoV-2) which is a novel enveloped *Betacoronavirus*, a member of the *Coronaviridae* family with a positive sense single stranded RNA genome [3, 4]. The common clinical manifestations include fever, cough, fatigue, sore throat, headache which in severe cases may progress to acute respiratory distress syndrome, cytokine storm, multiple organ failure and death [3, 5–8].

The SARS-CoV-2 is known to cause manifestations in organ systems including the gastrointestinal tract and ocular tissues [9, 10]. While the main route of transmission is via respiratory droplets [11, 12], studies conducted during the SARS-CoV pandemic caused by a virus which is phylogenetically similar to SARS-CoV-2 did show the presence of viral RNA in tear samples [13, 14]. During the SARS-associated coronavirus outbreak of 2003, one study found that the most predictive variable for transmission of the infection from infected patients to healthcare workers was whether or not the healthcare workers used protective eyewear [7]. This raised questions about potential alternative modes of transmission. During the current pandemic, there have been various reports of ocular involvement including features of follicular conjunctivitis in patients infected with SARS-CoV 2 with some of them even demonstrating the presence of viral RNA in conjunctival or tear specimens collected from these patients [13–16]. However, the implications of ocular involvement in the course and prognosis of the systemic disease are not known.

Data compilation is needed to determine the occurrence and nature of ocular manifestations associated with COVID-19 and the percentage of cases of reverse transcriptase polymerase chain reaction (RT-PCR) positivity for viral RNA in ocular fluids. We performed a systematic review and meta-analysis to evaluate the epidemiology and clinical features of ocular surface manifestations and complications related to COVID-19 infection, to assess the risk of transmission to patients and healthcare workers through ocular secretions and to determine whether ocular involvement has any correlation with the severity of respiratory symptoms and overall prognosis.

## Materials and methods

The study protocol investigating the proportion of patients with ocular conditions in COVID-19 can be found at PROSPERO (registration number: CRD42020182623). The study was performed in accordance to the PRISMA guidelines [17]. The PRISMA checklist is provided in S1 Appendix.

## Inclusion and exclusion criteria

Both prospective and retrospective studies and case series reporting ophthalmic manifestations in confirmed COVID-19 patients of any age group (children and adults), either gender, any race/ethnicity were included in the study. Confirmed COVID-19 cases indicate patients who were diagnosed COVID-19 positive either on the basis of clinical criteria (as recommended by their National Health agencies) [18] or positive RT-PCR for viral RNA from nasopharyngeal swabs. An attempt was made to obtain unpublished literature as well. Two authors decided upon the inclusion of studies, and 2 others performed quality assessment. Discrepancies, if any, were resolved by discussion.

Studies with only suspected cases of COVID-19 were excluded from analysis. Case reports, letter to editors (not reporting cases), narrative reviews, and correspondence (such as editorials) were also excluded.

## Literature search

A systematic literature search of electronic databases i.e. PubMed, Web of Science, CINAHL, OVID and Google scholar was performed by 2 independent reviewers The searches included literature from December 1, 2019 through May 31, 2020. Publications in English language, or those which had English language translation were included in the analysis. The search strategy is provided in S2 Appendix.

## Data collection and analysis

A pre-piloted structured form was used to extract data from the included studies about study setting, study design, demographic details of patients, occurrence of various ocular symptoms and complications, status of RT-PCR positivity from tear and/or conjunctival samples, systemic disease status and prognosis of the patients. Two reviewers extracted the data independently and if any discrepancies arose, they were settled after discussion with third reviewer who acted as an arbiter.

We used STATA MP 2 Core to perform the meta-analysis where possible. One reviewer entered the data and another performed a crosscheck for at least 20% of the entered data for correctness. If any discrepancies were found, the data was re-entered by a third reviewer. We pooled the data from similar studies using the inverse variance & random effects method. The pooled data was reported as effect estimates (percentages with 95% CI). We conducted a narrative synthesis of the studies if meta-analysis was not feasible. $I^2$ statistics were used for investigating heterogeneity and the following interpretation of $I^2$ was applied [19]:

- 0% to 40%: might not be important;

- 30% to 60%: may represent moderate heterogeneity;

- 50% to 90%: may represent substantial heterogeneity;

- 75% to 100%: considerable heterogeneity.

Galbraith plot was used to represent the heterogeneity of studies. Risk of bias assessment was performed based on the quality assessment checklist for prevalence study by Hoy *et al.* which has a high inter-rater agreement [20]. This checklist assesses the risk of bias on various domains including representativeness of the target population, random selection, likelihood of non-response bias, data collection, use of case definitions, reliability and validity of measuring tools, and appropriate use of numerator and denominator for the ocular symptoms. If the

criteria was fulfilled, individual items were rated as "yes", and scored 0. At the end, the total numeric score was obtained by adding the responses which were "no". The risk of bias was low if the numeric score was 0–3, moderate if the score was 4–6, and high risk if the score was 7–9. In order to analyse publication bias, Begg's test, Egger's linear regression and the inverted funnel plots were used.

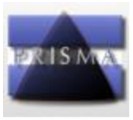 **PRISMA 2009 Flow Diagram**

**Identification**

Records identified through database searching
(n = 222)

**Screening**

Records after duplicates removed
(n = 109)

Records screened
(n = 109)

Records excluded
(n = 76)

**Eligibility**

Full-text articles assessed for eligibility
(n = 33)

Full-text articles excluded, with reasons
(n = 17)

Please write the reasons for exclusion like below

6 review articles

5 outcomes were different

**Included**

Studies included in qualitative synthesis
(n = 16)

Studies included in quantitative synthesis (meta-analysis)
(n = 8)

**Fig 1. The PRISMA flow chart is represented in Fig 1.**

## Study outcomes

The primary study outcomes included:

1. The proportion of patients with ocular involvement in confirmed cases of COVID-19.

**Table 1. The characteristics of the studies included in the systematic review.**

| Sr. No | Author | Study Design, Setting | Location | Month, Year | Study Population | Mean age (years), % males | Method of ophthalmic data collection | Ocular fluid analysis |
|---|---|---|---|---|---|---|---|---|
| 1 | Hong et al. [21] | Cross-sectional | China | March 2020 | Hospitalized patients (isolation ward) | 48, 55.4 | Questionnaire | None |
| | | Hospital setting | | | | | Subjective recall | |
| 2 | Zhang et al. [22] | Cross-sectional | China | February 2020 | Patients (including healthcare workers) | 57.6, 47.1 | Ocular examination | Conjunctival swabs |
| | | Hospital setting | | | | | | |
| 3 | Tostmann et al. [23] | Cross-sectional | Netherlands | March 2020 | Healthcare workers | NA, 21.1 | Questionnaire | None |
| | | Hospital setting | | | | | | |
| 4 | Wu et al. [14] | Cross-sectional | China | March 2020 | Hospitalized patients | 68, 65.8 | Ocular examination | Conjunctival swabs |
| | | Hospital setting | | | | | | |
| 5 | Zhou et al. [24] | Cross-sectional | China | February 2020 | Hospitalized patients | 35.7, 22.2 | Questionnaire or interview | Conjunctival swabs |
| | | Hospital setting | | | | | | |
| 6 | Lan et al. [25] | Cross-sectional | China | April 2020 | Hospitalized patients | 41.6, 40.7 | Ocular examination | Conjunctival swabs |
| | | Hospital setting | | | | | | |
| 7 | Xu et al. [26] | Cross-sectional | China | April 2020 | Hospitalized patients | 43.7, 53.3 | Ocular examination | Conjunctival swabs |
| | | Hospital setting | | | | | | |
| 8 | Karimi et al. [27] | Cross-sectional | Iran | May 2020 | Hospitalized patients | 56.6, 67.4 | Ocular examination | Conjunctival swabs |
| | | Hospital setting | | | | | | |
| 9 | Chen et al. [28] | Prospective case series | China | March 2020 | Hospitalized patients | 40 and 50*, 50.2 | Telephonic interview | None |
| | | Hospital setting | | | | | Questionnaire | |
| 10 | Seah et al. [29] | Prospective case series | Singapore | March 2020 | Patients | 37**, 65 | Ocular examination | Tear samples |
| | | Hospital setting | | | | | | |
| 11 | Xia et al. [13] | Prospective case series | China | February 2020 | Hospitalized patients | 54.5, 70 | Ocular examination | Conjunctival swabs |
| | | Hospital setting | | | | | | |
| 12 | Scalinci et al. [30] | Prospective case series | Italy | April 2020 | Eye hospital patients | 46.8, 80 | Ocular examination | None |
| | | Hospital setting | | | | | | |
| 13 | Guan et al. [31] | Retrospective | China | April 2020 | Patients (both hospitalized and outpatient) | 47**, 58.1 | Medical records | None |
| | | Hospital setting | | | | | | |
| 14 | Marinho et al. [32] | Prospective case series | Brazil | May 2020 | Healthcare workers | NA, 50 | Ocular examination | None |
| | | Hospital setting | | | | | Optical coherence tomography | |
| 15 | Xie et al. [33] | Prospective case series | China | April 2020 | Patients | 57.6, 66.7 | Ocular examination | Conjunctival swabs |
| | | Hospital setting | | | | | | |
| 16 | Deng et al. [34] | Prospective case series | China | April 2020 | Hospitalized patients (including those in intensive care) | 61.4, 54.4 | No ocular examination performed | Conjunctival swabs |
| | | Hospital setting | | | | | | |

* The manuscript reports two cohorts from different hospitals, and has reported median age of the subjects separately.

** Indicates median age (not mean age).

Table 2. Risk of bias for individual studies included in the meta-analysis as per Hoy et al. [20]*.

| Risk of bias domains | Hong et al. [21] | Zhang et al. [22] | Tostmann et al. [23] | Wu et al. [14] | Zhou et al. [24] | Lan et al. [25] | Xu et al. [26] | Karimi et al. [27] |
|---|---|---|---|---|---|---|---|---|
| 1. Was the study's target population a close representation of the national population in relation to relevant variables eg: age, sex | 1 | 1 | 1 | 1 | 1 | 1 | 1 | 1 |
| 2. Was the sampling frame a true or close representation of the target population? | 1 | 0 | 0 | 0 | 0 | 0 | 1 | 1 |
| 3. Was some form of random selection used to select the sample, OR, was a census undertaken? | 1 | 1 | 1 | 1 | 1 | 1 | 1 | 1 |
| 4. Was the likelihood of non-response bias minimal? | 0 | 0 | 1 | 0 | 0 | 0 | 0 | 0 |
| 5. Were data collected directly from the subjects (as opposed to a proxy)? | 0 | 0 | 0 | 0 | 0 | 0 | 0 | 0 |
| 6. Was an acceptable case definition used in the study? | 0 | 1 | 1 | 0 | 0 | 1 | 0 | 0 |
| 7. Was the study instrument that measured the parameter of interest shown to have reliability and validity? | 1 | 0 | 0 | 1 | 1 | 1 | 1 | 0 |
| 8. Was the same mode of data collection used for all subjects? | 0 | 0 | 0 | 0 | 0 | 0 | 0 | 0 |
| 9. Were the numerator(s) and denominator(s) for the parameter of interest appropriate? | 1 | 1 | 1 | 1 | 1 | 1 | 1 | 1 |
| Summary on the overall risk of study bias | 5 | 4 | 5 | 4 | 4 | 5 | 5 | 4 |

Score of 0 indicates low risk; 1 indicates high risk.

* The risk of bias assessment is indicated by the following: score 0–3 is low risk; 4–6 is moderate risk, and 7–9 is high risk.

2. Clinical features, demographic profile and ocular complications of COVID-19 patients.

3. Percentage of patients with COVID-19 whose first clinical manifestation was in the form of ocular involvement.

Secondary outcomes of the study included:

1. Systemic profile, disease severity, and survival outcomes of patients diagnosed with COVID-19 with ocular disease.

2. RT-PCR positivity from conjunctival/tear samples of confirmed cases of COVID-19.

## Results

We identified 222 citations through the search of electronic databases. After removing the duplicate articles, screening of titles and abstracts was performed for 109 articles. Full text screening was done for 33 manuscripts of which 16 fulfilled the inclusion criteria and were used for data extraction. The PRISMA flowchart shows the full screening process (Fig 1). The characteristics of the included studies are listed in Table 1. A total of 3064 patients were reported in the included studies of which 2347 were confirmed cases of COVID-19. Overall, 196 patients (8.35%) were reported to have ocular surface manifestations.

### Risk of bias in the included studies

The risk of bias assessment revealed that all 8 studies included in the meta-analysis exhibited moderate risk of bias [14, 21–27] (Table 2). None of the studies were deemed to have a high risk of bias.

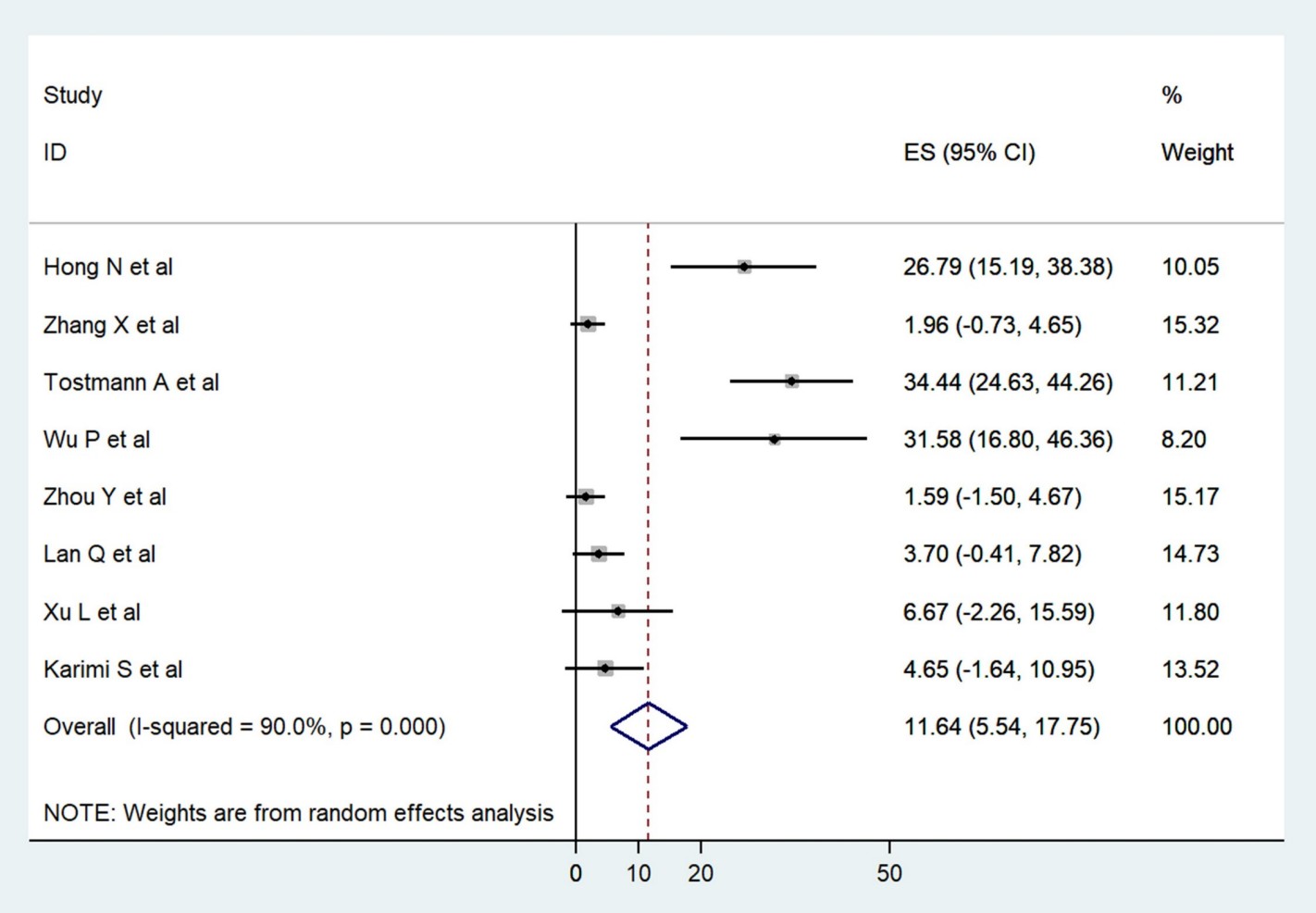

**Fig 2. Forest plot showing the proportion of ocular symptoms reported from cross-sectional studies on COVID-19 patients.**

## Proportion of ocular manifestations

Pooled analysis using random effects model from 8 cross-sectional observational studies showed that 11.64% (95%CI: 5.54–17.75) of COVID-19 patients had some form of ocular symptoms (Fig 2) [14, 21–27]. Pooled data from 6 studies showed ocular manifestations in the form of follicular conjunctivitis in 7% patients (95%CI: 2.12–11.89) [14, 21, 22, 24, 25, 27]. Three studies also reported conjunctival chemosis in 4.44% patients (95%CI: -1.13–10.01) [14, 22, 25] while ocular redness or conjunctival congestion was present in 10.89% patients (95%CI: 3.04–18.74) [14, 21, 22, 24]. The details of these 8 studies for which meta-analysis was performed are provided in Table 3. The presence of ocular features including discharge, itching, pain, and watering is listed in Fig 3. The other uncommon ocular symptoms are also provided in Fig 3.

Ocular manifestations were also reported in the case series not included in the pooled analysis [13, 28–34]. Chen *et al.* reported various ocular signs and symptoms in their cohort of 534 subjects with COVID-19 [28]. Seah *et al.* [29] and Xia *et al.* [13] in their prospective case series reported one case each of conjunctival chemosis and follicular conjunctivitis, respectively.

**Table 3. The prevalence of ocular symptoms in patients with COVID-19 included in the pooled analysis.**

| Sr. No | Author | Total COVID-19 Patients | Total patients with ocular symptoms | Symptoms (number of eyes) |
|---|---|---|---|---|
| 1 | Hong at al [21] | 56 | 15 | Redness (15) |
| | | | | Dryness (15) |
| | | | | Ocular pain (15) |
| | | | | Foreign body sensation (15) |
| | | | | Discharge (15) |
| | | | | Itching (15) |
| | | | | Follicular conjunctivitis (15) |
| 2 | Zhang et al. [22] | 112 | 2 | Watering (1) |
| | | | | Redness (1) |
| | | | | Conjunctival chemosis (1) |
| | | | | Follicular conjunctivitis (2) |
| 3 | Tostmann et al. [23] | 90 | 31 | Ocular Pain (31) |
| 4 | Wu et al. [14] | 38 | 12 | Watering (12) |
| | | | | Redness (12) |
| | | | | Discharge (12) |
| | | | | Conjunctival chemosis (12) |
| | | | | Follicular conjunctivitis (12) |
| 5 | Zhou et al. [24] | 63 | 1 | Redness (1) |
| | | | | Discharge (1) |
| | | | | Itching (1) |
| | | | | Follicular conjunctivitis (1) |
| 6 | Lan et al. [25] | 81 | 3 | Dryness (1) |
| | | | | Conjunctival chemosis (1) |
| | | | | Swelling (2) |
| | | | | Itching (3) |
| | | | | Follicular conjunctivitis (3) |
| 7 | Xu et al. [26] | 30 | 2 | Itching (1) |
| | | | | Macular degeneration (1)* |
| 8 | Karimi et al. [27] | 43 | 2 | Foreign body sensation (1) |
| | | | | Follicular conjunctivitis (1) |

* one patient in this study had pre-existing macular degeneration.

Scalinci *et al.* [30] have reported a series of 5 patients whose sole clinical manifestation of COVID-19 was acute follicular conjunctivitis. These studies are summarized in Table 4.

## Ocular manifestations as first symptom of COVID-19

Pooled data analysis from 5 studies showed that ocular symptoms were the first manifestation in 2.26% (95%CI: 0.03–4.49) of patients (Fig 4) [14, 21, 24, 25, 28]. In addition to the studies included in the meta-analysis, Scalinci *et al.* [30] reported 5 cases where acute follicular conjunctivitis was the first and sole manifestation of COVID-19.

## Severe pneumonia in patients with ocular symptoms

Three prospective cross-sectional studies reported the occurrence of severe pneumonia in patients with ocular involvement [14, 22, 33]. Analysis of pooled data revealed that 6.91%

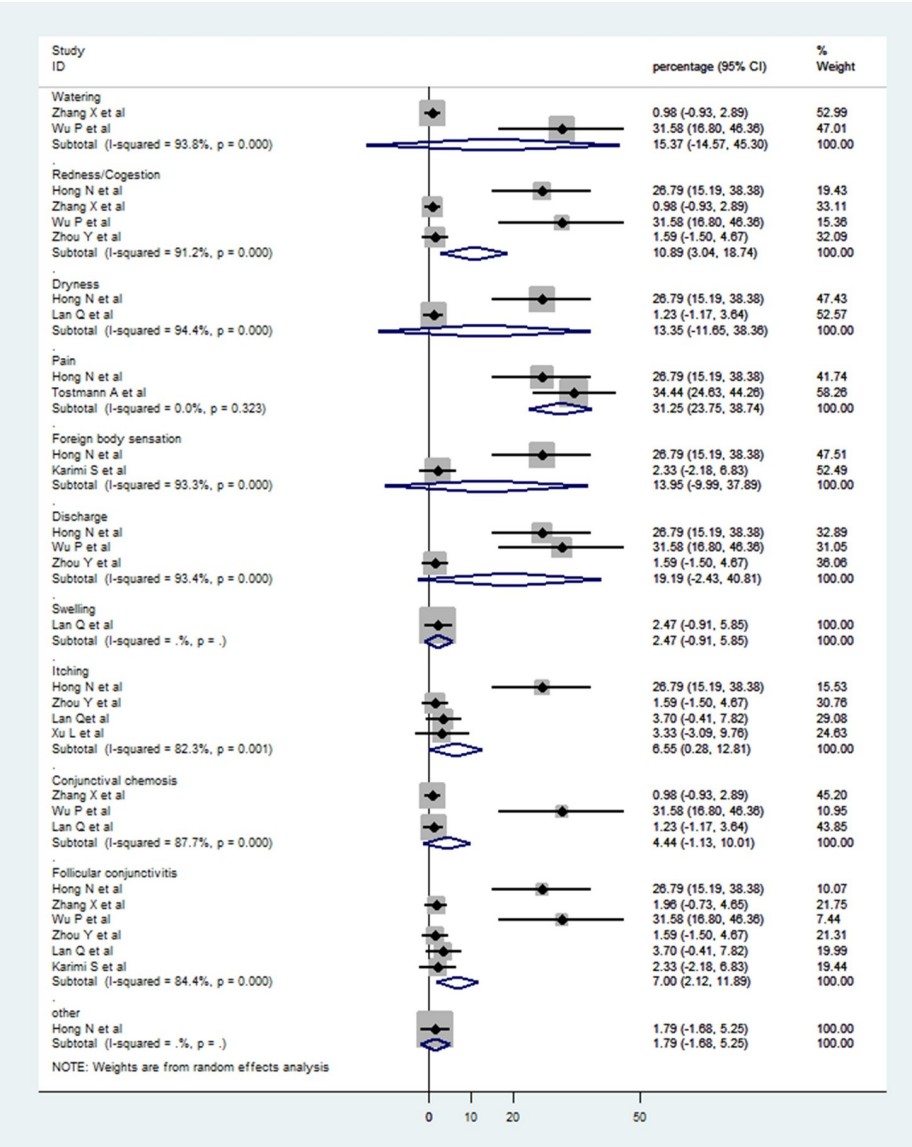

**Fig 3. Forest plot showing subgroup analysis for various ocular symptoms amongst patients of COVID-19 who had ocular manifestations.**

(95% CI: -1.75–15.58) of the patients with ocular manifestations suffered from severe pneumonia. Wu *et al.* [14] reported that ocular manifestations were more common among patients with severe pneumonia. In their series, patients with ocular manifestations had higher mean white blood cell counts, absolute neutrophil counts, C-reactive protein, D-dimer levels, and lactate dehydrogenase. Patients with severe pneumonia had blood saturation <93%, dyspnea, shock or multiple organ failure. Zhang *et al.* [22] and Xie *et al.* [33] reported one patient each with ocular manifestations and severe pneumonia. Xie *et al.* [33] reported a 90-year-old patient with fever, dyspnea and headache who succumbed to multi-organ failure. Zhang *et al.* [22] described severe pneumonia in a 29-year-old nurse with multiple peripheral ground glass opacities in both lungs on computerized chest

**Table 4. The prevalence of ocular symptoms in patients with COVID-19 not included in the pooled analysis.**

| Sr. No | Author | Total COVID-19 Patients | Total patients with ocular symptoms | Symptoms (number of eyes) |
|---|---|---|---|---|
| 1 | Chen at al [28] | 534 | 112 | Dryness (112) |
| | | | | Blurring (68) |
| | | | | Foreign body sensation (63) |
| | | | | Watering (55) |
| | | | | Discharge (52) |
| | | | | Itching (52) |
| | | | | Follicular conjunctivitis (33) |
| | | | | Redness (25) |
| | | | | Ocular pain (22) |
| | | | | Photophobia (15) |
| | | | | Marginal keratitis (14) |
| 2 | Seah et al. [29] | 17 | 1 | Conjunctival chemosis (1) |
| 3 | Xia et al. [13] | 30 | 1 | Discharge (1) |
| | | | | Follicular conjunctivitis (1) |
| 4 | Scalinci et al. [30] | 5 | 5 | Watering (5) |
| | | | | Redness (5) |
| | | | | Photophobia (5) Discharge (5) Chemosis (5) |
| | | | | Follicular conjunctivitis (5) |
| 5 | Guan et al. [31] | 1099 | 9 | Redness (9) |
| 6 | Marinho et al. [32] | 12 | 0 | - |
| 7 | Xie et al. [33] | 33 | 0 | - |
| 8 | Deng et al. [34] | 114 | 0 | - |

tomography, and elevated white cell counts. Four studies reported mild to moderate respiratory symptoms in 4.13% patients (95% CI -0.31–8.56) with ocular symptoms (Fig 5) [14, 24, 26, 32].

## RT-PCR positivity in ocular fluids of COVID-19 patients

Six studies provided data on RT-PCR positivity from conjunctival swabs or tear samples of COVID-19 patients with or without presence of ocular signs and symptoms [13, 14, 21, 22, 24, 27]. These six studies reported a total of 335 patients out of which 12 (3.5%) had RT-PCR positive results. Thus, viral RNA was detected in 3.5% (95% CI 0.87–6.13) of COVID-19 patients from ocular samples collected on single or multiple occasions. In addition, Xia *et al.* [13] reported one case of COVID-19 with acute follicular conjunctivitis who was RT-PCR positive (Fig 6).

## Heterogeneity and publication bias

Asymmetrical inverted funnel plot showed significant publication bias despite a comprehensive and exhaustive search for the studies (Fig 7). Egger's linear regression showed a significant small study effect and confirmed the significant publication bias in the meta-analysis (Fig 8). Galbraith's plot (Fig 9) showed that three studies (Hong *et al.* [21], Tostmann *et al.* [23] and Wu *et al.* [14]) were a source of significant heterogeneity as they had higher patients with ocular symptoms compared to other studies. This could be attributed to the study design by Hong *et al.* [21] and Tostmann *et al.* [23], which employed a

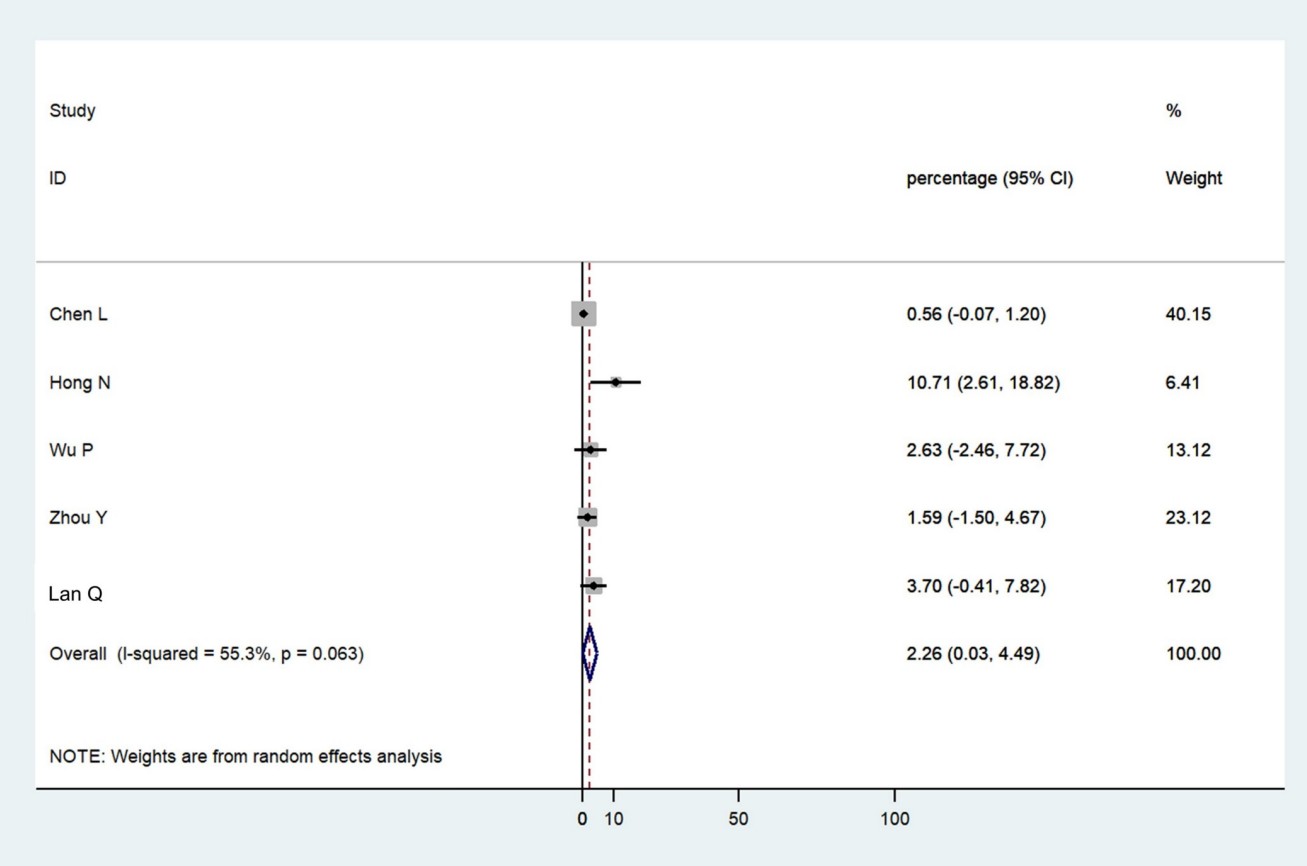

**Fig 4. Forest plot showing the proportion of patients who had ocular manifestations as the first symptom of COVID-19.**

detailed questionnaire. Tostmann *et al.* [23] included only healthcare workers as their study sample. Wu *et al.* [14] did not provide details on the methods of data collection and ocular examination.

## Discussion

During the ongoing COVID-19 pandemic, there have been several manuscripts published in the literature regarding various ocular manifestations of the disease. In our systematic review, we observed that the overall percentage of the ocular manifestations was approximately 11% from the meta-analysis of studies. The major ophthalmic features reported with COVID-19 were ocular pain, redness, and follicular conjunctivitis. Since some of these reports were published in early 2020 when the WHO had declared COVID-19 as a pandemic, there is a significant concern regarding the extrapulmonary manifestations of COVID-19 and risk of transmission of the disease through ocular fluids. Several authors have published recommendations on the use of protective eye gear to avoid potential transmission of the disease. These recommendations include strategies to prevent transmission of the disease among ophthalmologists and contact lens practitioners, and from aerosols generated from ocular procedures such as cataract surgery and non-contact tonometry [35–42].

The data from eight studies included in the pooled analysis revealed a significant proportion of various ocular features, specifically ocular pain, redness, discharge and follicular

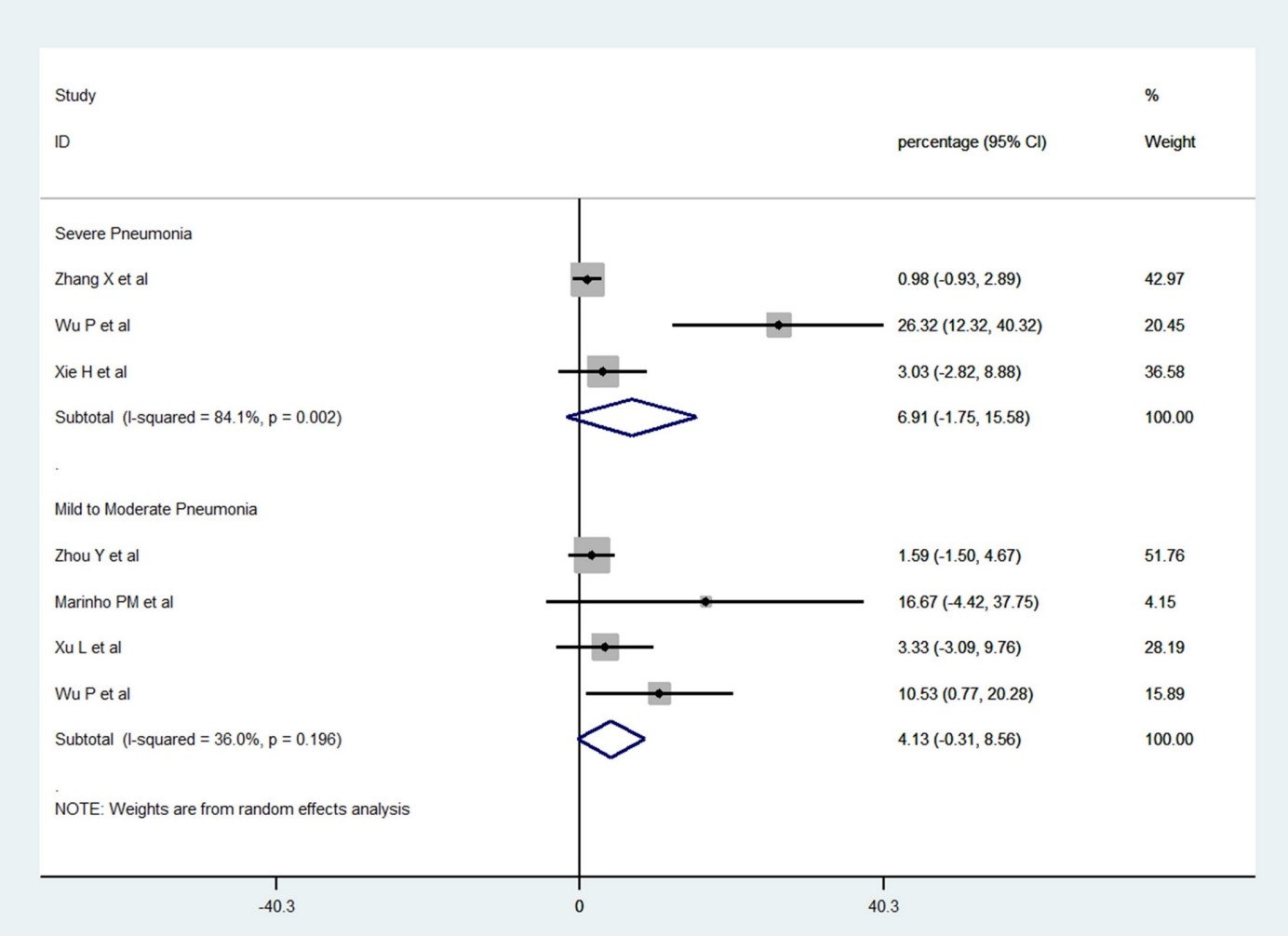

**Fig 5. Forest plot showing the proportion of COVID-19 patients with ocular manifestations who had severe or mild to moderate pneumonia.**

conjunctivitis [14, 21–27]. Other studies also reported similar ocular manifestations. However, it must be noted that several studies relied on detailed and exhaustive questionnaires and patient interviews, which were performed several days after the patients were discharged from the hospital/recovered [21, 23, 24, 28, 31]. Therefore, the data from the studies could suffer from recall bias. In addition, it is not clear whether these ocular features were pre-existing or occurred as a result of COVID-19 infection. For instance, features such as dry eyes, itching and foreign body sensation may be highly prevalent in the general population given the high incidence of dry eye disease [43–45]. Moreover, certain studies have included healthcare workers who may be more sensitized on reporting various symptoms [22, 23, 32]. Healthcare workers do not represent the general population and this must be considered while interpreting the results of these studies. On the other hand, in life-threatening situations, the more severe clinical manifestations may take precedence over ophthalmic features, which may go unnoticed [46].

The studies reporting ocular manifestations lack several critical details. Due to the high risk of disease transmission, direct slit-lamp examination was not performed by a majority of the

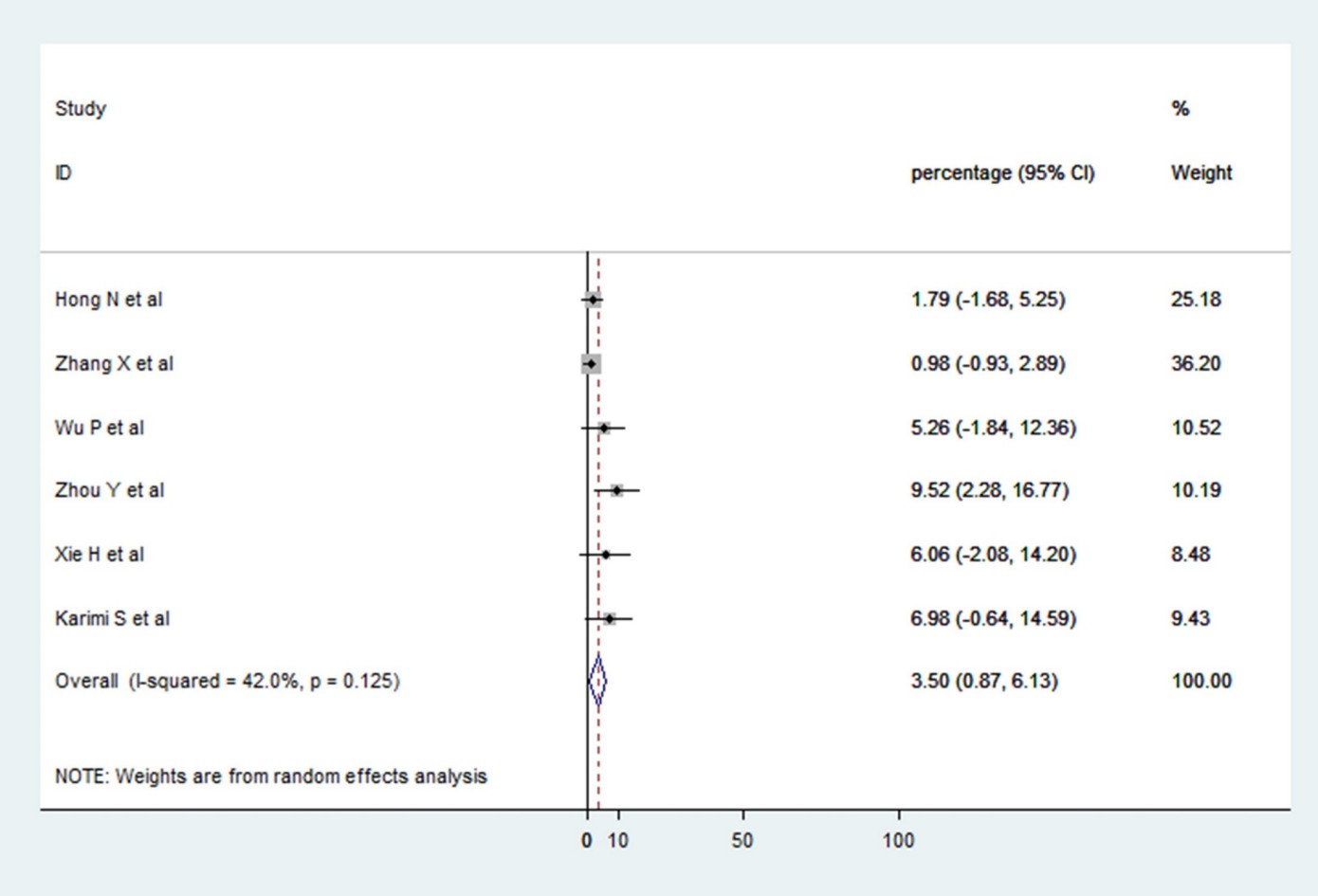

**Fig 6. Forest plot showing the proportion of COVID-19 patients who had positive reverse transcriptase polymerase chain reaction (RT-PCR) from conjunctival/tear samples.**

authors. The characteristics of the ocular pain such as location, nature, and duration, features of ocular discharge, extent and severity of redness/congestion (involvement of palpebral/bulbar conjunctiva or circumcorneal congestion), and adnexal features of follicular conjunctivitis do not have detailed descriptions. Since only one study has reported optical coherence tomography features including hyper-reflectivity of retinal layers, and micro-hemorrhages in the fundus, the relevance of these findings is unknown [32]. The studies reporting ocular features also do not provide any information on potential drugs used to control these manifestations. There are number of antiviral agents such as remdesivir, favipiravir, and galidesivir under consideration against coronavirus [47]. Preclinical studies are also evaluating the role of other compounds such as non-anticoagulant sulphated polysaccharides [48] against the virus, as these agents have the advantage of local mucosal delivery.

Non-specificity of the ocular manifestations is another concern in the meta-analysis. Due to morbidities such as severe pneumonia and intensive care admission, several patients may develop non-specific conditions such as dry eyes, pain, chemosis and redness [49–51]. These features may not be directly related to the underlying disease, and may be observed in severely ill subjects. The available data does not permit any concrete conclusions in this regard.

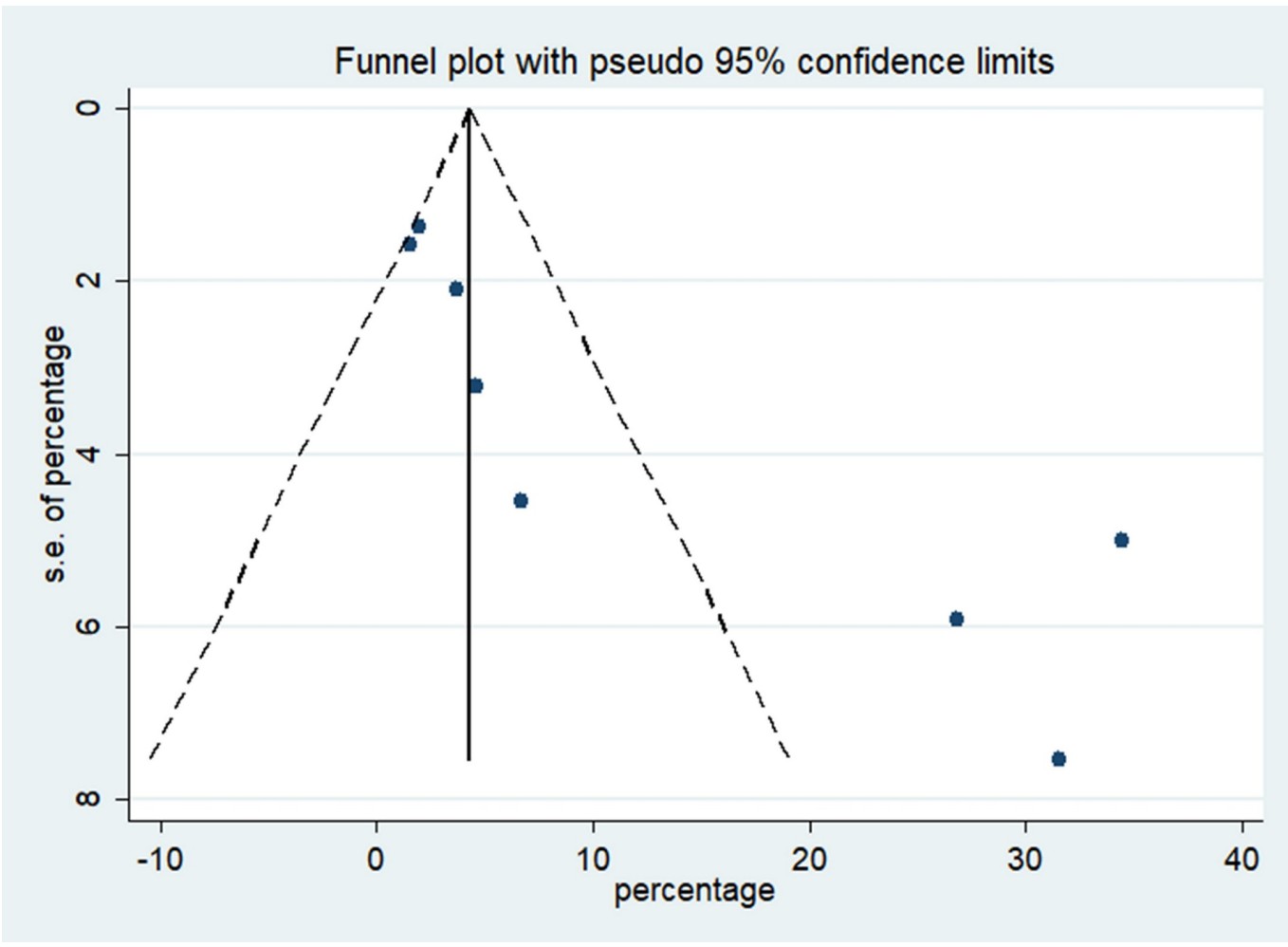

**Fig 7. Inverse funnel plot showing the publication bias of the cross-sectional studies included in the meta-analysis.**

The overall positivity from ocular fluids of SARS-CoV-2 RNA was rather uncommon. In certain studies, conjunctival swabs did not reveal any RNA from the studied cohort [29, 34]. It is unclear if the viral RNA present in the ocular fluids has infectious potential, or has resulted in actual disease transmission thus far. Thus, the risk of transmission of the disease from ocular fluids may have been overestimated in the literature. Nonetheless, it is prudent to remain cautious and consider the risk of transmission till robust data to negate this possibility is published. The relative lack of detectable viral RNA in the ocular fluids may also raise another question–whether the ocular manifestations of COVID-19 are truly due to the viral infection of ocular tissues, or are they a spectrum of the flu-related ocular symptoms accompanying several viral illnesses [52]. The virus has not been cultured from ocular fluids so far. In addition, one study investigating the cytopathic effect of the virus on Vero-E6 cell lines failed to demonstrate such changes on cell lines [29]. Previous studies have shown the utility of Vero-E6 cell lines in isolating herpes simplex virus, and studying its cytopathic effect in the context of herpes simplex keratitis [53].

The published studies also raise a concern that COVID-19 can have ocular manifestations (specifically follicular conjunctivitis) as the first and sometimes the sole manifestation of the

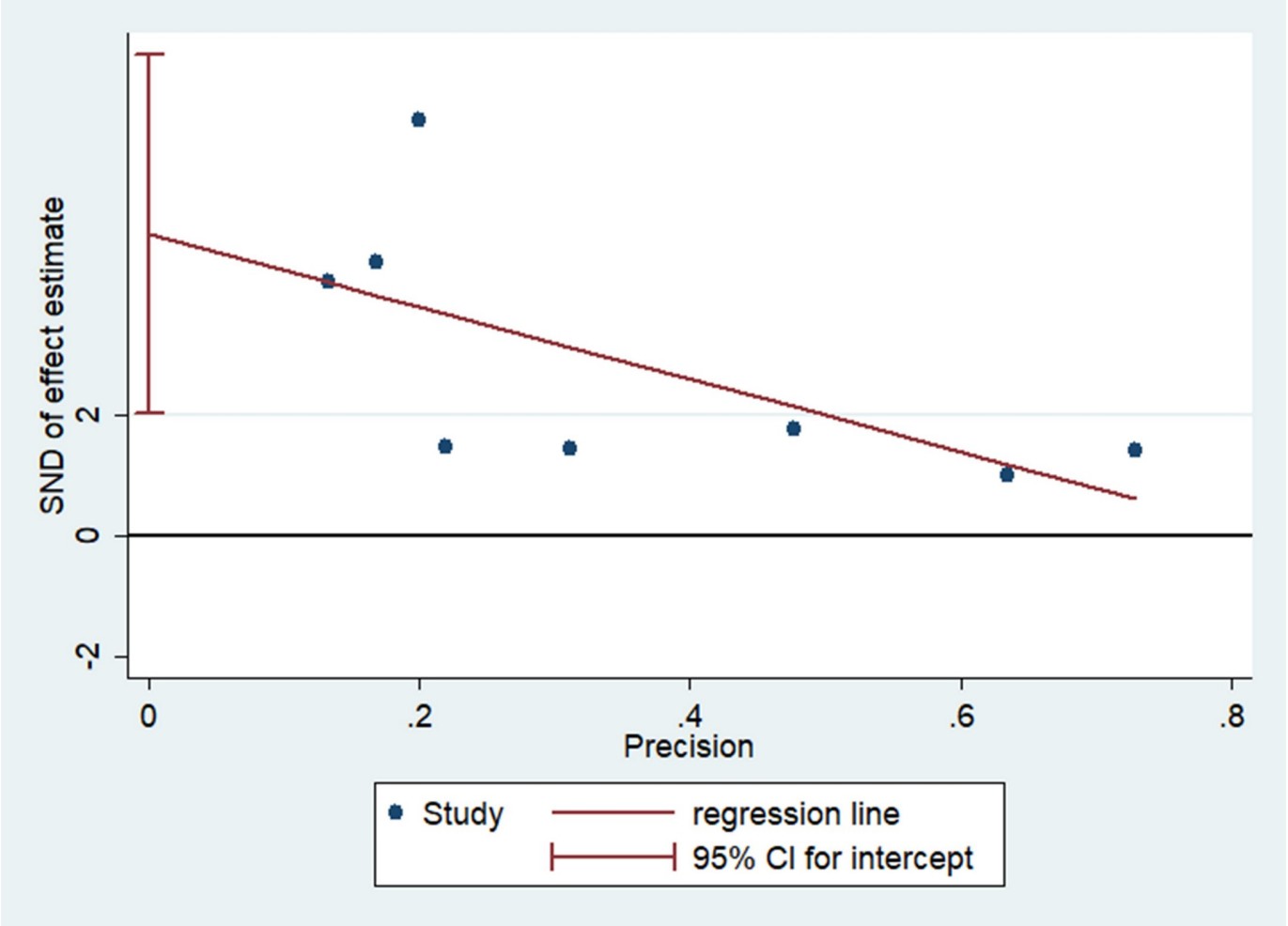

**Fig 8. Egger's linear regression showed a significant publication bias in the meta-analysis.**

disease. Pooled data from three studies in our meta-analysis revealed that ocular symptoms may be the first manifestation in approximately 2.2% patients only [14, 21, 24, 25, 28]. As ophthalmologists, it is important to be aware of such presentations and keep a high index of clinical suspicion of COVID-19 in such patients. In addition, analysis of pooled data revealed that 6.91% of the patients with ocular manifestations suffered from severe pneumonia [14, 22, 33]. A hypercoagulable state causing arterial and venous thromboembolic complications has been described in severe cases of COVID-19 [31, 54, 55]. However, none of the studies described any features related to ocular thrombotic complications.

A major limitation of the published data is the heterogeneity of certain studies due to higher proportion of patients with ocular symptoms compared to others [14, 21, 23]. This could be attributed to detailed questionnaires used in these studies. Such study designs suffer from significant bias in their methodology. Despite an exhaustive search, publication bias was also reported in our meta-analysis. Certain manuscripts included in the systematic review were pre-prints available on online databases and had not been peer-reviewed [24, 26, 28]. We excluded case reports in our meta-analysis, some of which have reported ocular manifestations of COVID-19. Thus, lack of uniformity in collecting and reporting of data is a major limitation of the published studies.

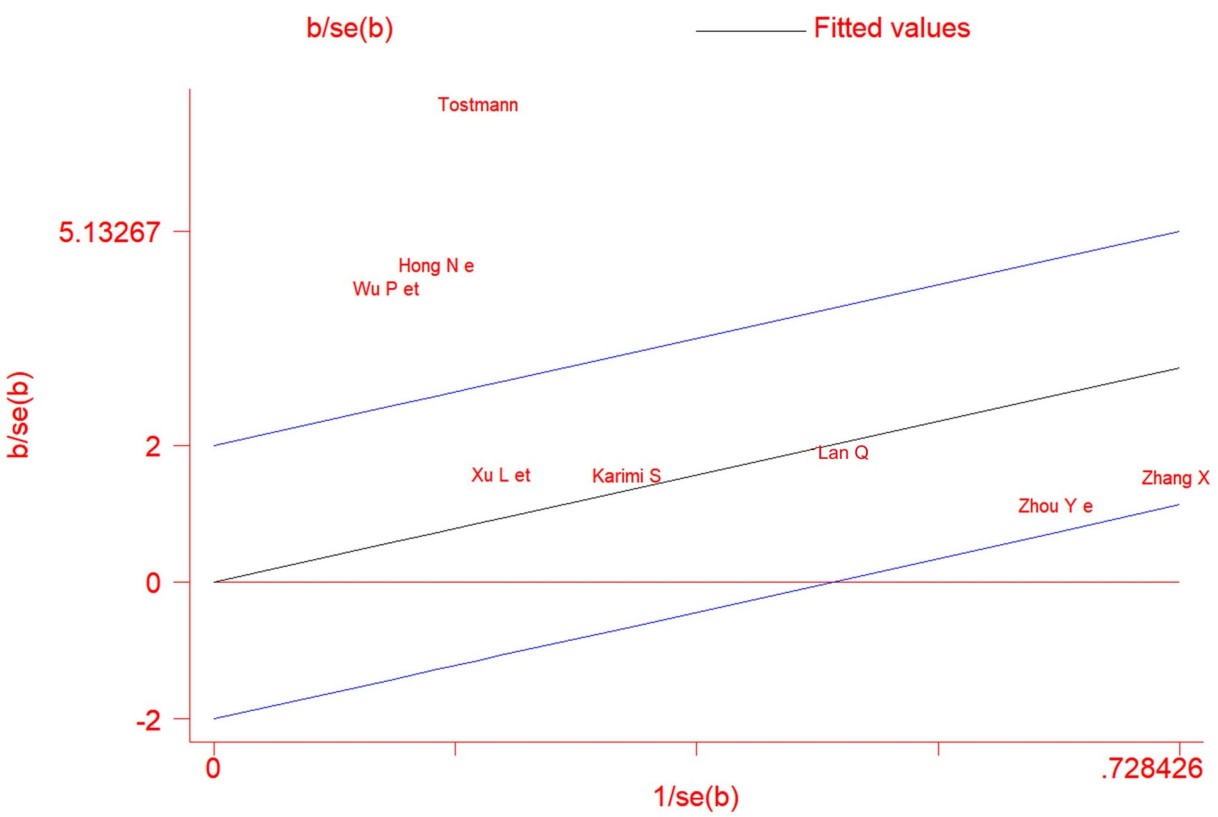

**Fig 9. Galbraith plot showing heterogeneity of the cross-sectional studies included in the meta-analysis.**

In summary, ocular manifestations such as pain, redness and conjunctivitis may be observed in subjects with COVID-19. The transmissibility of the disease from ocular fluids remains uncertain, and the rates of viral RNA detection from conjunctival swabs/tear fluid using RT-PCR are low. Such a systematic analysis may aid planning agencies, ophthalmologists, and intensivists in managing their patients, and in developing guidelines on personal protective equipment including eye gear. In the future, robust data collection, analysis and reporting is desirable so that there is better understanding of the risk of ocular transmission, and the overall prevalence of the ocular disease in COVID-19.

## Supporting information

**S1 Appendix. The Preferred Reporting Items for Systematic Reviews and Meta-Analyses (PRISMA) checklist has been provided.**
(DOC)

**S2 Appendix. Search strategy used for the systematic review.**
(DOCX)

## Acknowledgments

ICMR Centre for Advanced Research in Evidence Based Child Health, PGIMER for providing guidance for literature search and software access for analysis.

## Author Contributions

**Conceptualization:** Kanika Aggarwal, Aniruddha Agarwal, Rupesh Agrawal, Vishali Gupta.

**Data curation:** Kanika Aggarwal, Aniruddha Agarwal, Nishant Jaiswal, Neha Dahiya, Alka Ahuja, Louis Tong.

**Formal analysis:** Kanika Aggarwal, Aniruddha Agarwal, Nishant Jaiswal, Neha Dahiya, Alka Ahuja, Louis Tong, Vishali Gupta.

**Methodology:** Kanika Aggarwal, Aniruddha Agarwal, Nishant Jaiswal, Neha Dahiya, Alka Ahuja, Sarakshi Mahajan, Louis Tong, Mona Duggal, Meenu Singh, Rupesh Agrawal, Vishali Gupta.

**Project administration:** Kanika Aggarwal, Aniruddha Agarwal, Mona Duggal, Meenu Singh, Rupesh Agrawal, Vishali Gupta.

**Resources:** Nishant Jaiswal.

**Supervision:** Sarakshi Mahajan, Louis Tong, Mona Duggal, Meenu Singh, Rupesh Agrawal, Vishali Gupta.

**Validation:** Kanika Aggarwal.

**Writing – original draft:** Kanika Aggarwal, Aniruddha Agarwal, Nishant Jaiswal, Neha Dahiya, Alka Ahuja, Sarakshi Mahajan.

**Writing – review & editing:** Kanika Aggarwal, Nishant Jaiswal, Neha Dahiya, Alka Ahuja, Sarakshi Mahajan, Louis Tong, Mona Duggal, Meenu Singh, Rupesh Agrawal, Vishali Gupta.

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
