## [Decision Letter · Decision Letter 0]

8 Sep 2020

PONE-D-20-22384

Ocular surface manifestations of coronavirus disease 2019 (COVID-19): a systematic review and meta-analysis

PLOS ONE

Dear Dr. Gupta,

Thank you for submitting your manuscript to PLOS ONE. After careful consideration, we feel that it has merit but does not fully meet PLOS ONE’s publication criteria as it currently stands. Therefore, we invite you to submit a revised version of the manuscript that addresses the points raised during the review process. Please make sure that you address all the concerns raised by the reviewers.

Please submit your revised manuscript in 60 days. If you will need more time than this to complete your revisions, please reply to this message or contact the journal office at plosone@plos.org. Please include the following items when submitting your revised manuscript:

We look forward to receiving your revised manuscript.

Kind regards,

Deepak Shukla

Academic Editor

PLOS ONE

Journal Requirements:

2. We note that you assessed risk of bias with a checklist with domains including representativeness of the target population, random selection, likelihood of non-response bias, data collection, use of case definitions, reliability and validity of measuring tools, and appropriate use of numerator and denominator for the ocular symptoms. Please ensure that you have provided a table in your manuscript with the scores for all individual risk of bias domains for all studies included your systematic review.

Reviewers' comments:

Reviewer's Responses to Questions

**Comments to the Author**

1. Is the manuscript technically sound, and do the data support the conclusions?

Reviewer #1: Yes

Reviewer #2: Yes

2. Has the statistical analysis been performed appropriately and rigorously? 

Reviewer #1: Yes

Reviewer #2: I Don't Know

3. Have the authors made all data underlying the findings in their manuscript fully available?

Reviewer #1: Yes

Reviewer #2: Yes

4. Is the manuscript presented in an intelligible fashion and written in standard English?

Reviewer #1: Yes

Reviewer #2: Yes

5. Review Comments to the Author

Reviewer #1: This is an excellent meta-analysis by the authors discussing considerable amount of data available out there. While the data that they have used is not perfect (due to heterogeneity of testing methods in the original studies), the authors have tried to extract as much meaningful data as possible. The authors were able to remove as many non-compliant studies as possible from their manuscript as possible while providing good meaningful data.

Minor revisions

1. May be the authors could talk about some potential drugs that were used in these studies to control the ocular manifestation. Some info can be found in PMID: 32714563, PMID: 32560227

2. Page 14 first line, "white" needs to be "while"

3. The authors say that 3.5% of the patients had RT-PCR based viral RNA detection in ocular samples. It wasnt very clear in the manuscript whether this percentage is out of 3000 patients or 2300 patients or only the studies that did conjuctival swabs. It would be good to know the total number.

4. Figure 2 and 3, quality of the image is not very good. May be provide a better version for publication.

Reviewer #2: Ocular surface manifestations of coronavirus disease 2019 (COVID-19): a systematic review and meta-analysis

Reviewer’s comments

In this study, the authors have described the meta-analysis of cases having ocular manifestation of COVID-19 around the world. The manuscript highlights the meta-analysis of clinical symptoms during ocular manifestation of COVID-19 such as ocular pain, discharge, redness and follicular conjunctivitis. The meta-analysis has been performed in which 222 citations were identified by online database search. After screening process described by the authors, 16 papers were found relevant. The data from 16 research articles was used which were published from China, Netherlands, Iran, Singapore, Italy and Brazil. The data includes a sum total of 2347 positive cases out of 3064 patients. Out of these, 196 patients (8.35%) were reported with ocular manifestation.

The meta-analysis does not include any studies published from India.

Also, the meta-analysis of pooled data revealed that 6.91% of the patients with ocular manifestations suffered from severe pneumonia the association of ocular manifestation with the severity of the respiratory complications or the prognosis is not discussed in details.

Incidence refers to the occurrence of new cases of disease or injury in a population over a specified period of time.

Abstract: Prevalence: Prevalence in epidemiology is the proportion of a particular population found to be affected by a medical condition at a specific time.

Incidence refers to the occurrence of new cases of disease or injury in a population over a specified period of time.

But the population included in this study is not uniform. So instead of prevalence, incidence/occurrence can be a better word.

● The study states that 11.6% patients had ocular manifestation of COVID-19. Out of these Ocular pain (31.2%), discharge (19.2%), redness (10.8%), and follicular conjunctivitis (7.7%) were the main features. But 31.2 + 19.2 + 10.8 + 7.7 = 68.9 %.

● Does remaining 31.1% patients signify some other uncommon symptoms? Please specify in details.

Introduction

● The studies published thus far suffer from significant publication bias and heterogeneity, and robust data collection and reporting of ocular manifestations is needed in the future. (Please re-phrase this sentence)

● It was declared a public health emergency of international concern on 30th January 2020 and in March 2020, the World Health Organization (WHO) declared the COVID-19 outbreak a pandemic (Please rephrase this sentence: Outbreak and Pandemic are two different epidemiological terms).

● The causative pathogen of this potentially fatal disease is a novel enveloped RNA beta coronavirus 2, a member of the Coronaviridae family, also known as severe acute respiratory syndrome coronavirus2 (Please rephrase this sentence. Enveloped positive sense ss-RNA virus. Betacoronavirus and Coronaviridae is written in Italics)

● During the current pandemic, there have been various reports of ocular involvement including features of follicular conjunctivitis in patients infected with SARS-CoV 2 with some of them even demonstrating the presence of viral RNA in conjunctival or tear specimens collected from these patients.

● Rate of reverse transcriptase polymerase chain reaction (RT-PCR) positivity for viral RNA in ocular fluids (The rate of positivity can be calculated in a Cohort study. Because rate is with respect to time in which the study is been performed. In a cross-sectional study, only number of positive cases can be stated)

Materials and methods

● Confirmed COVID-19 cases indicate patients who were diagnosed COVID-19 positive either on the basis of clinical criteria or positive RT-PCR for viral RNA from nasopharyngeal swabs (Please check about this line). (The diagnosis should be done on the basis of clinical signs and confirmation by RT-PCR. Only clinical criteria might not be sufficient)

● The heterogeneity on the basis of I2:

• 0% to 40%: might not be important;

• 30% to 60%: may represent moderate heterogeneity;

• 50% to 90%: may represent substantial heterogeneity;

• 75% to 100%: considerable heterogeneity.

Please provide a reference it might be needed.

Study outcome:

● Clinical features, demographic profile and ocular complications of COVID-19 patients with ophthalmic manifestation (Ocular and ophthalmic is repetition: might need to rephrase the sentence)

Risk of bias: The risk of bias is demonstrated to be moderate in this study. But it needs more elaboration. Also, in the discussion part questions have been raised by the authors regarding risk of bias. This might contradict the study.

Table 1:

● The reference Zhang et al (Ref no 24) is actually Xu et al., 2020 (Please check author names)

● (Reference number should be checked properly: The reference numbers given in the table are interchanged)

● Tostmann et al: (Ref no 21): 20 number

● Zhou et al: (Ref no 22): 21 number

● Qianqian et al., (Ref no 23) is actually Lan et al., 2020

● Table 1: row number 7 reference not cited properly

Discussion

Several questions have been raised in the discussion part of the review. Possibilities of various bias that can be encountered during the data collections have been discussed such as recall bias, study design bias and bias due to pre-existing ocular features. Apart from this, few more constraints such as in severe cases ocular features may go unnoticed or non-specific conditions such as dry eyes, pain chemosis and redness. These conditions might subject the judgement of healthcare workers to be biased. Therefore, in many cases the conjunctival swabs or tear samples did not test positive for SARS-CoV-2 RNA by qRT-PCR.

But the data presented in this study shows, only 2.2% of the patients show ocular manifestation as the first clinical sign. Apart from this 6.9% patients with ocular symptoms suffered from severe pneumonia. This data is not sufficient to have a concrete conclusion. More data analysis might be considered for the analysis. Also, the analysis should be supported by significant statistical analysis.

6. PLOS authors have the option to publish the peer review history of their article (what does this mean?). If published, this will include your full peer review and any attached files.

Reviewer #1: No

Reviewer #2: **Yes: **Dr Shyam Sundar Nandi

---

## [Author Response · Author response to Decision Letter 0]

16 Sep 2020

Dr. Deepak Shukla

Academic Editor

PLOS One

Ref: PONE-D-20-22384 Ocular surface manifestations of coronavirus disease 2019 (COVID-19): a systematic review and meta-analysis

Thank you very much for the comment. We have ensured that the manuscript complies with the journal style requirements.

2. We note that you assessed risk of bias with a checklist with domains including representativeness of the target population, random selection, likelihood of non-response bias, data collection, use of case definitions, reliability and validity of measuring tools, and appropriate use of numerator and denominator for the ocular symptoms. Please ensure that you have provided a table in your manuscript with the scores for all individual risk of bias domains for all studies included your systematic review.

We have provided a new table detailing the scores for all the individual risk of bias domains for all the studies for which meta-analysis was performed.

New Table 2

Thank you very much for letting us know. We have provided captions for the supporting information files and updated the in-text citations.

Reviewer 1:

1. This is an excellent meta-analysis by the authors discussing considerable amount of data available out there. While the data that they have used is not perfect (due to heterogeneity of testing methods in the original studies), the authors have tried to extract as much meaningful data as possible. The authors were able to remove as many non-compliant studies as possible from their manuscript as possible while providing good meaningful data.

Thank you very much for the detailed review and constructive suggestions. We have modified the manuscript based on the comments. We hope these changes are satisfactory, and the manuscript is now acceptable for publication.

2. May be the authors could talk about some potential drugs that were used in these studies to control the ocular manifestation. Some info can be found in PMID: 32714563, PMID: 32560227

None of the studies included in the meta-analysis have mentioned any details related to 

treatment given for ocular signs and symptoms such as follicular conjunctivitis. However, based on the reviewer’s suggestion, we have extracted information from the two suggested articles and added to the discussion.

Page 15, 1st Paragraph

3. Page 14 first line, "white" needs to be "while" 

We have made the correction. Thank you very much.

Page 14, First Line

4. The authors say that 3.5% of the patients had RT-PCR based viral RNA detection in ocular samples. It wasn’t very clear in the manuscript whether this percentage is out of 3000 patients or 2300 patients or only the studies that did conjunctival swabs. It would be good to know the total number.

As mentioned in the manuscript, only six studies provided data on RT-PCR positivity from conjunctival swabs or tear samples of COVID-19 patients. These 6 studies reported a total of 335 patients and out of these 12 patients had RT-PCR positive from ocular fluids/swabs irrespective of presence or absence of ocular symptoms. The pooled analysis of data from these studies revealed that 3.5% of COVID-19 patients were RT-PCR positive from ocular samples. We have added the total number of patients whose tear samples or conjunctival swabs were tested by RT-PCR in the manuscript text. 

Page 12, Last Paragraph

5. Figure 2 and 3, quality of the image is not very good. May be provide a better version for publication.

We have improved the quality of all the figures in the revised version of the manuscript. Thank you very much for the suggestion. 

All Figures

Reviewer 2:

1. In this study, the authors have described the meta-analysis of cases having ocular manifestation of COVID-19 around the world. The manuscript highlights the meta-analysis of clinical symptoms during ocular manifestation of COVID-19 such as ocular pain, discharge, redness and follicular conjunctivitis. The meta-analysis has been performed in which 222 citations were identified by online database search. After screening process described by the authors, 16 papers were found relevant. The data from 16 research articles was used which were published from China, Netherlands, Iran, Singapore, Italy and Brazil. The data includes a sum total of 2347 positive cases out of 3064 patients. Out of these, 196 patients (8.35%) were reported with ocular manifestation. The meta-analysis does not include any studies published from India. Also, the meta-analysis of pooled data revealed that 6.91% of the patients with ocular manifestations suffered from severe pneumonia. The association of ocular manifestation with the severity of the respiratory complications or the prognosis is not discussed in details.

Thank you very much for a thorough review of our study and for your constructive suggestions. We have incorporated all the suggestions in order to improve the manuscript. The aim of this analysis was to determine whether patients of COVID-19 who have ocular manifestations have a more severe systemic disease in the form of severe pneumonia or other systemic complications and whether they were associated with a poorer prognosis. Only 3 studies reported severe pneumonia in patients with ocular symptoms. Wu P et al found that ocular features were present more commonly in patients with severe pneumonia and were associated with higher white blood cell and neutrophil counts and higher levels of procalcitonin, C-reactive protein, and lactate dehydrogenase than patients without ocular symptoms. Similarly, the descriptions by Xie et al and Zhang et al have been added to the revised manuscript. Thank you very much. 

Page 12, First Paragraph

2. Incidence refers to the occurrence of new cases of disease or injury in a population over a specified period of time. Abstract: Prevalence: Prevalence in epidemiology is the proportion of a particular population found to be affected by a medical condition at a specific time. Incidence refers to the occurrence of new cases of disease or injury in a population over a specified period of time. But the population included in this study is not uniform. So instead of prevalence, incidence/occurrence can be a better word.

Thank you very much for the input. We have replaced ‘prevalence’ with “occurrence” or “proportion” in the abstract and everywhere else in the manuscript.

3. The study states that 11.6% patients had ocular manifestation of COVID-19. Out of these ocular pain (31.2%), discharge (19.2%), redness (10.8%), and follicular conjunctivitis (7.7%) were the main features. But 31.2 + 19.2 + 10.8 + 7.7 = 68.9%.

Does remaining 31.1% patients signify some other uncommon symptoms? Please specify in details.

Thank you very much. The ocular manifestations such as ocular pain, discharge, redness and follicular conjunctivitis were the most common ones. Other uncommon symptoms included - watering, dryness, foreign body sensation, itching, swelling, and conjunctival chemosis. The word limit in the abstract does not allow us to add these. These have been provided in Figure 3 along with the studies reporting them. This has been indicated in the revised manuscript in the results section. 

Abstract

4. The studies published thus far suffer from significant publication bias and heterogeneity, and robust data collection and reporting of ocular manifestations is needed in the future. (Please re-phrase this sentence).

We have re-phrased this sentence as “The available studies show significant publication bias and heterogeneity. Prospective studies with methodical collection and data reporting are needed for evaluation of ocular involvement in COVID-19.” 

Abstract

5. It was declared a public health emergency of international concern on 30th January 2020 and in March 2020, the World Health Organization (WHO) declared the COVID-19 outbreak a pandemic (Please rephrase this sentence: Outbreak and Pandemic are two different epidemiological terms).

We have re-phrased this sentence as “It was declared a public health emergency of international concern on 30th January 2020 and in March 2020, it was labelled as a pandemic by the World Health Organization (WHO).” 

Page 5, First Paragraph, Introduction

6. The causative pathogen of this potentially fatal disease is a novel enveloped RNA beta coronavirus 2, a member of the Coronaviridae family, also known as severe acute respiratory syndrome coronavirus2 (Please rephrase this sentence. Enveloped positive sense ss-RNA virus. Betacoronavirus and Coronaviridae is written in Italics)

Thank you very much. We have re-phrased this sentence as “The causative pathogen of this potentially fatal disease has been named severe acute respiratory syndrome coronavirus2 (SARS-CoV-2) which is a novel enveloped Betacoronavirus, a member of the Coronaviridae family with a positive sense single stranded RNA genome.” 

Page 5, First Paragraph, Introduction

7. Rate of reverse transcriptase polymerase chain reaction (RT-PCR) positivity for viral RNA in ocular fluids (The rate of positivity can be calculated in a Cohort study. Because rate is with respect to time in which the study is been performed. In a cross-sectional study, only number of positive cases can be stated)

Thank you very much. We have replaced ‘rate’ with ‘percentage of cases’.

Page 6, Second Paragraph

8. Confirmed COVID-19 cases indicate patients who were diagnosed COVID-19 positive either on the basis of clinical criteria or positive RT-PCR for viral RNA from nasopharyngeal swabs (Please check about this line). (The diagnosis should be done on the basis of clinical signs and confirmation by RT-PCR. Only clinical criteria might not be sufficient)

Thank you very much. We have checked all included studies and according to the currently available data, majority of the studies included laboratory/RT-PCR confirmed cases of COVID-19 for studying ocular manifestations or tear/conjunctival sampling. However, a few studies, apart from including RT-PCR confirmed COVID-19 patients, have also mentioned including clinically confirmed cases of COVID-19 based on their respective national guidelines such as the National Health Commission in the studies from China. The updated reference has been added.

Page 7, First Paragraph

9. The heterogeneity on the basis of I2:

• 0% to 40%: might not be important;

• 30% to 60%: may represent moderate heterogeneity;

• 50% to 90%: may represent substantial heterogeneity;

• 75% to 100%: considerable heterogeneity.

Please provide a reference it might be needed.

Thank you very much. We have included the reference for this in the manuscript.

Deeks JJ, Higgins JPT, Altman DG (editors). Chapter 10: Analysing data and undertaking meta-analyses. In: Higgins JPT, Thomas J, Chandler J, Cumpston M, Li T, Page MJ, Welch VA (editors). Cochrane Handbook for Systematic Reviews of Interventions version 6.0 (updated July 2019). Cochrane, 2019. Available from www.training.cochrane.org/handbook.

Page 8, New reference 19

10. Clinical features, demographic profile and ocular complications of COVID-19 patients with ophthalmic manifestation (Ocular and ophthalmic is repetition: might need to rephrase the sentence)

Thank you very much. We have modified this study outcome as “Clinical features, demographic profile and ocular complications of COVID-19 patients”.

Page 9, Study Outcomes, bullet point 2

11. Risk of bias: The risk of bias is demonstrated to be moderate in this study. But it needs more elaboration

We agree with the reviewer that the risk of bias needs further explanation. To elaborate on this, we have now included a table in the manuscript with the scores for all individual risk of bias domains for all studies included in our systematic review. Thank you very much for this suggestion.

New Table 2

12. Table 1: The reference Zhang et al (Ref no 24) is actually Xu et al., 2020 (Please check author names). (Reference number should be checked properly: The reference numbers given in the table are interchanged). Tostmann et al: (Ref no 21): 20 number. Zhou et al: (Ref no 22): 21 number. Qianqian et al., (Ref no 23) is actually Lan et al., 2020. Table 1: row number 7 reference not cited properly

Thank you very much for highlighting this shortcoming. We have corrected all the references and checked the author names. The reference Qianqian et al has been corrected to Lan et al. 

13. Several questions have been raised in the discussion part of the review. Possibilities of various bias that can be encountered during the data collections have been discussed such as recall bias, study design bias and bias due to pre-existing ocular features. Apart from this, few more constraints such as in severe cases ocular features may go unnoticed or non-specific conditions such as dry eyes, pain chemosis and redness. These conditions might subject the judgement of healthcare workers to be biased. Therefore, in many cases the conjunctival swabs or tear samples did not test positive for SARS-CoV-2 RNA by qRT-PCR. But the data presented in this study shows, only 2.2% of the patients show ocular manifestation as the first clinical sign. Apart from this 6.9% patients with ocular symptoms suffered from severe pneumonia. This data is not sufficient to have a concrete conclusion. More data analysis might be considered for the analysis.

Thank you very much for this suggestion. Since there are a number of limitations of the published studies, we have highlighted these limitations in the discussion section. We have mentioned in the revised manuscript that the available data is not sufficient to arrive at a concrete conclusion. Further analysis would not be possible with this dataset. Thank you once again for a thorough review of our manuscript. 

General Comment

PLOS authors have the option to publish the peer review history of their article (what does this mean?). If published, this will include your full peer review and any attached files.

Thank you very much for providing us the option. We would like to go ahead with publishing the peer review history of our article.

---

## [Decision Letter · Decision Letter 1]

20 Oct 2020

Ocular surface manifestations of coronavirus disease 2019 (COVID-19): a systematic review and meta-analysis

PONE-D-20-22384R1

Dear Dr. Gupta,

We’re pleased to inform you that your manuscript has been judged scientifically suitable for publication and will be formally accepted for publication once it meets all outstanding technical requirements.

Kind regards,

Deepak Shukla

Academic Editor

PLOS ONE

Additional Editor Comments (optional):

Reviewers' comments:

Reviewer's Responses to Questions

**Comments to the Author**

1. If the authors have adequately addressed your comments raised in a previous round of review and you feel that this manuscript is now acceptable for publication, you may indicate that here to bypass the “Comments to the Author” section, enter your conflict of interest statement in the “Confidential to Editor” section, and submit your "Accept" recommendation.

Reviewer #1: All comments have been addressed

Reviewer #2: All comments have been addressed

2. Is the manuscript technically sound, and do the data support the conclusions?

Reviewer #1: Yes

Reviewer #2: Yes

3. Has the statistical analysis been performed appropriately and rigorously? 

Reviewer #1: Yes

Reviewer #2: Yes

4. Have the authors made all data underlying the findings in their manuscript fully available?

Reviewer #1: Yes

Reviewer #2: Yes

5. Is the manuscript presented in an intelligible fashion and written in standard English?

Reviewer #1: Yes

Reviewer #2: Yes

6. Review Comments to the Author

Reviewer #1: (No Response)

Reviewer #2: The authors have adequately addressed the comments previously made. All the queries are answered satisfactorily. The article should be accepted for publication.

7. PLOS authors have the option to publish the peer review history of their article (what does this mean?). If published, this will include your full peer review and any attached files.

Reviewer #1: No

Reviewer #2: **Yes: **Dr Shyam Sundar Nandi

---

## [Editor Report · Acceptance letter]

26 Oct 2020

PONE-D-20-22384R1 

Ocular surface manifestations of coronavirus disease 2019 (COVID-19): a systematic review and meta-analysis 

Dear Dr. Gupta:

I'm pleased to inform you that your manuscript has been deemed suitable for publication in PLOS ONE. Congratulations! Your manuscript is now with our production department. 

Kind regards, 

on behalf of

Prof. Deepak Shukla 

Academic Editor

PLOS ONE